# OpenReview forum: "Zer0-Jack: A memory-efficient gradient-based jailbreaking method for black box Multi-modal Large Language Models"
_NeurIPS.cc/2024/Workshop/SafeGenAi — SafeGenAi Poster_

### Official Review · Reviewer_pvig · 2024-10-10
**The proposed method is almost the same as SPSA, then it raises the question of whether introducing a new name for this method is warranted.**

**Rating:** 5
**Confidence:** 5

**Review:**

This paper proposes a black-box attack against multi-modal large language models (LLMs). The proposed method, Zer0-Jack, leverages zeroth-order optimization techniques to generate malicious images, rather than relying on perturbations derived from a surrogate white-box model with subsequent transfer attacks.

The approach is notable, as accessing internal information from LLMs is significantly more challenging than from traditional deep neural networks (DNNs), making black-box settings more practical and promising. Furthermore, the method demonstrates effective attacks using limited memory resources, which is a valuable advantage in resource-constrained environments.

However, a key concern arises regarding the similarity of the proposed method in Eq. (4) to the Simultaneous Perturbation Stochastic Approximation (SPSA) algorithm [1][2]. Since the core technique—estimating the gradient of a black-box model—is fundamentally the same as SPSA, it raises the question of whether introducing a new name for this method is warranted. If the proposed method does not significantly deviate from SPSA in terms of its underlying principles or algorithmic innovations, it may be more appropriate to acknowledge it as an extension or application of SPSA rather than branding it as a new technique.


[1]Uesato, Jonathan, et al. "Adversarial risk and the dangers of evaluating against weak attacks." International conference on machine learning. PMLR, 2018.

[2]James C Spall. Multivariate stochastic approximation using a simultaneous perturbation gradient approximation. IEEE transactions on automatic control, 37(3):332–341, 1992.

---

### Official Review · Reviewer_3gxk · 2024-10-11
**Blackbox jailbreak method for MLLM**

**Rating:** 5
**Confidence:** 5

**Review:**

This paper proposes a black-box method for jailbraking MLLM with zero-th order optimization method.
The paper has poor quality in clarity and organization. It was difficult for me to follow the discussion. The abstract is way too long w.r.t. intro. Intro lacks of introduction to the problem and the problem statement itself. Related works section is limited so a few paper on the matter but many other things are cited in the rest of the paper. Moreover, (and finally) a bit of problem definition is presented in the Method Section.
Overview of the method and Figure 3 should be in the body of the paper and not an appendix.
Moreover the setup of the LLM is not described and it is of utter importance when dealing with "alignment".
I was a bit confused when authors talk about black-box and commercial LLMs. This sentence: "Zer0-Jack achieves a 95% attack success rate on MiniGPT-4 with the Harmful Behaviors Multi-modal Dataset, demonstrating its effectiveness. Additionally, we show that Zer0-Jack can directly attack commercial MLLMs such as GPT-4o"... MiniGPT-4 is not commercial?

The method proposed seems novel and effective but I don't have data to evaluate the correctness of the results presented given that supplementary material and code is not available for me now.

In conclusion, this paper can be improved in writing and clarity. The method has novelty and provides effectiveness in the obtained results. However at review time I cannot assess further the correctness of the evaluation technique and, thus, the presented results.